# Multi-Indicator and Geospatial Based Approaches for Assessing Variation of Land Quality in Arid Agroecosystems

Ahmed S Abuzaid [1], Yasser S. A. Mazrou [2,3], Ahmed A El Baroudy [4], Zheli Ding [5] and Mohamed S. Shokr [4,*]

1 Soils and Water Department, Faculty of Agriculture, Benha University, Benha 13518, Egypt; ahmed.abuzaid@fagr.bu.edu.eg
2 Applied College—Muhyle, King Khalid University, Abha 62587, Saudi Arabia; yasser.mazroua@agr.tanta.edu.eg
3 Department of Agriculture Economic, Faculty of Agriculture, Tanta University, Tanta 31527, Egypt
4 Soil and Water Department, Faculty of Agriculture, Tanta University, Tanta 31527, Egypt; drbaroudy@agr.tanta.edu.eg
5 Haikou Experimental Station, Chinese Academy of Tropical Agricultural Sciences, Haikou 570000, China; dingzheli@zju.edu.com
* Correspondence: mohamed_shokr@agr.tanta.edu.eg

**Abstract:** Novel spatial models for appraising arable land resources using data processing techniques can increase insight into agroecosystem services. Hence, the principal component analysis (PCA), hierarchal cluster analysis (HCA), analytical hierarchy process (AHP), fuzzy logic, and geographic information system (GIS) were integrated to zone and map agricultural land quality in an arid desert area (Matrouh Governorate, Egypt). Satellite imageries, field surveys, and soil analyses were employed to define eighteen indicators for terrain, soil, and vegetation qualities, which were then reduced through PCA to a minimum data set (MDS). The original and MDS were weighted by AHP through experts' opinions. Within GIS, the raster layers were generated, standardized using fuzzy membership functions (linear and non-linear), and assembled using arithmetic mean and weighted sum algorithms to produce eight land quality index maps. The soil properties (pH, salinity, organic matter, and sand), slope, surface roughness, and vegetation could adequately express the land quality. Accordingly, the HCA could classify the area into eight spatial zones with significant heterogeneity. Selecting salt-tolerant crops, applying leaching fraction, adopting sulfur and organic applications, performing land leveling, and using micro-irrigation are the most recommended practices. Highly significant ($p < 0.01$) positive correlations occurred among all the developed indices. Nevertheless, the coefficient of variation (CV) and sensitivity index (SI) confirmed the better performance of the index developed from the non-linearly scored MDS and weighted sum model. It could achieve the highest discrimination in land qualities (CV > 35%) and was the most sensitive (SI = 3.88) to potential changes. The MDS within this index could sufficiently represent TDS ($R^2 = 0.88$ and Kappa statistics = 0.62), reducing time, effort, and cost for estimating the land performance. The proposed approach would provide guidelines for sustainable land-use planning in the studied area and similar regions.

**Keywords:** GIS; fuzzy logic; multivariate statistical analysis; AHP; land quality index

## 1. Introduction

Drylands are located in dry sub-humid, semi-arid, arid, and hyper-arid regions, where the aridity index (the ratio between annual rainfall and potential evapotranspiration) is below 0.65. They occupy 45.4% of the Earth's land surface and support more than 2 billion people around the world [1]. However, dryland agroecosystems undergo harsh climate conditions and poor land resources that negatively affect sustainable crop production [2]. The cultivated lands are vulnerable to degradation due to salinity, alkalinity, low organic matter, poor water quality, and sparse vegetation cover [3,4]. Hence, precise assessment

and monitoring of land qualities are essential for efficient land use planning and securing arable land resources [5,6].

Biological land resources are highly affected by climate, soil, terrain, and vegetation attributes [2]. Thus, land resources assessment is a multi-indicator evaluation process as numerous data are analyzed [7]. This entails selecting key indicators to build a minimum data set (MDS) to eliminate redundant data [8]. The most common methods used for this goal are expert opinion (EO) and statistical methods like principal component analysis (PCA) [9]. The EO is biased due to individual judgment, while PCA is more objective for reducing environmental data [10,11]. The PCA provides few combinations of un-correlated indicators and simplifies complex data sets without disturbing the original structure [12]. The efficiency of PCA in defining key indicators for dryland agroecosystems has been reported in previous studies on Farafra Oasis, Egypt [13], the Upper Tigris Basin [14], and Iran [15].

The indicators involved in assessing land resources are not equally effective in determining the ecosystem functions. Therefore, the relative importance (weight) of each parameter should be estimated [7]. The PCA has been used also as a multi-indicator weighting tool in studies related to natural resources assessment [11,16,17]. However, the number of studied cases is one of the main limitations since the PCA requires at least 150–300 cases [10,18]. When using a lower number of cases, the analytical hierarchal process (AHP) is the proper approach [16]. The AHP is a theory of measurement through pairwise comparisons based on EO to allocate a priority number within a 1 to 9 scale [19]. Thus, combined use of PCA as a data reduction tool and AHP weighting procedure can provide a better land resources assessment as reported for the Tigris Basin [14].

Remote sensing (RS) and geographic information system (GIS) are modern tools for assessing agroecosystems on different scales [2,3,5]. Satellite imageries provide a precise coverage of spatial data in a time-saving, reliable, and cost-effective manner, while GIS collect, edit, store, and display geo-referenced data [17]. The GIS-geostatistical analyst allows interpreting spatial variability of soil data and producing continuous layers to be included in zoning land resources [20]. The GIS statistical analyses tools are very important for evaluating variations in soil properties and predicting un-sampled locations [21]. The variogram analysis allows explaining the complex relations between soil data layers accurately [22]. The GIS-fuzzy membership functions (FMFs) convert raster layers to scores of 0 to 1, providing accurate handling with layer dimensions [7]. This reduces uncertainty, imprecision, and subjectivity related to manual methods [5]. The GIS-fuzzy and geostatistical tools have been efficiently employed to assess dryland agroecosystems in terms of crop suitability [23], land potentiality [5], and pasture soil quality [7].

The agroecosystems are viewed by spatio-temporal variability owing to various natural and human interventions affecting their services [24]. Hence, sustainable crop production entails delineating site-specific management zones to gather areas of similar properties and requirements in relatively homogenous zones [21]. Classification techniques such as hierarchical cluster analysis (HCA) assort data points into clusters or groups by performing a similarity test for these categories [25]. This, in turn, can adjust agronomic inputs (irrigation water and fertilizers) to optimize crop yield [26]. Thus, adopting precession agriculture practices through developing site-specific spatial land quality zones (SLQZs) would help in maintaining the function of agroecosystem functions.

In the drylands, limited studies focused on testing combinations of PCA, AHP, GIS-fuzzy sets to assess arable land resources and establish SLQZs through HCA. However, according to the aforementioned specifics, novel approaches based on integrations of these techniques would open new ways to conduct more rigorous and realistic simulations for dryland agroecosystem services. Hence, the main goals of this work are (i) applying PCA to opt key indicators affecting land performance, (ii) integrating PCA and HCA to delineate SLQZs, (iii) developing LQIs using geostatistical and fuzzy techniques within the GIS platform, and (iv) specifying the most appropriate index. The study was then applied in an

area typical for dryland agroecosystems in the Egyptian Western Desert for the upcoming evaluations in similar regions.

## 2. Materials and Methods

### 2.1. Study Area

The investigated area is one of the newly-developed zones in the Egyptian Western Desert, northeast Matrouh Governorate. It lies between 30°14′04″ to 30°20′43″ N and 28°47′48″ to 28°53′05″ E, covering a total area of 7945.76 ha (Figure 1). The elevation height ranges from −53 to 34 m above sea level. The area is dominated by arid climate conditions with an average temperature ranges from 6 °C (January) to 36 °C (July), and annual rainfall varies from 25 to 50 mm year$^{-1}$ [27]. Lower Miocene Moghra Formations (a thick layer of sand, silt, and clay mixed with minor carbonate interbeds) cover 98.5% of the total area, while Quaternary sand deposits cover 1.5% in the southern zone [28]. The soils are classified as Entisols and Aridisols. Of the 50 profiles, 37 represented *Typic Torripsamments*, 12 represented *Typic Torriorthents*, while one represented *Typic Haplosalids*. Only 238 ha (3%) is cultivated by olive and jojoba trees. The natural vegetation of halophyte species also covered scattered areas.

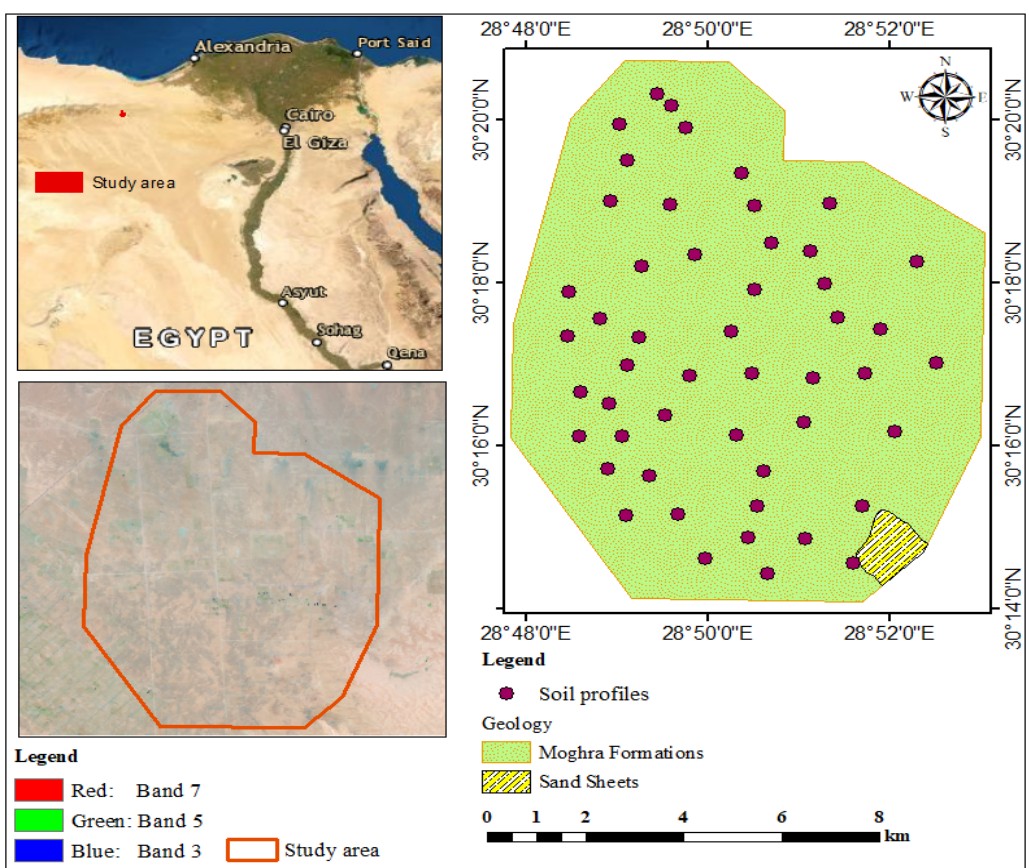

**Figure 1.** Location maps of the studied area.

### 2.2. Remote Sensing Data

One scene (path 178/row 39) of Landsat-8 (20-6-2021) and a digital elevation model (DEM) with a 30-m spatial resolution of the shuttle radar topographic mission (SRTM) were used. Within ArcGIS 10.8, the normalized difference vegetation index (NDVI) was calculated (Equation (1)).

$$NDVI = \frac{Band\ 5 - Band\ 4}{Band\ 5 + Band\ 4} \tag{1}$$

Terrain features were extracted from the DEM, including slope, aspect, topographic wetness index (TWI), surface roughness index (SRI), and slope length and steepness factor (LS) (Supplementary Figure S1). The TWI was calculated using Equation (2) [29], based on local slope contributing area ($A_s$) extracted from flow accumulation raster and the slope raster ($\beta$) as follows:

$$\text{TWI} = \text{Ln}\left(\frac{A_s}{\tan\beta}\right) \tag{2}$$

Focal statistics were applied to generate the SRI map using Equation (3) [30]:

$$\text{SRI} = \frac{\text{DEM}_{\text{mean}} - \text{DEM}_{\text{min}}}{\text{DEM}_{\text{max}} - \text{DEM}_{\text{min}}} \tag{3}$$

The LS factor was calculated using Equation (4) [31]:

$$\text{LS} = \left(\text{Flow accumulation} \times \frac{\text{Cell size}}{22.13}\right)^{0.4} \times \left(\frac{\text{Sin slope}}{0.0896}\right)^{1.3} \tag{4}$$

where flow accumulation expresses the contribution of an area accumulated upslope for a given cell, cell size is the size of the grid cell, and the sin slope is the slope degree value in sin.

### 2.3. Field Work and Laboratory Analyses

Fifty geo-referenced soil profiles (Figure 1) were dug to a 150 cm depth or lithic contact and were described [32]. The profile locations were chosen to represent the dominant geological formations in the studied area. Disturbed samples were collected from horizons, kept in polyethylene bags, and transferred to the laboratory. Undisturbed soil cores were taken to determine the bulk density (BD). The disturbed samples were air-dried, crushed, and sieved using a 2-mm mesh. Soil analyses were done according to Soil Survey Staff [33]. The particle size distribution (pipette method), water holding capacity (WHC), and hydraulic conductivity (HC) were determined. The pH and electrical conductivity (EC) were measured in 1:2.5 soil–water suspensions and in soil paste extracts, respectively. The organic matter (OM) was determined using the Walkley–Black procedure. The cation exchange capacity (CEC) and exchangeable sodium percentage (ESP) were determined (ammonium acetate at pH = 7.0). The $CaCO_3$ was determined using the calcimeter method. Main soil physicochemical properties are shown in Supplementary Table S1.

### 2.4. Modeling Land Quality

This involved five steps; (i) indicator selection, (ii) generating raster layers for soil attributes, (iii) thematic data standardization, (iv) weighting procedure, and (v) developing the final LQI maps (Supplementary Figure S2).

#### 2.4.1. Characterization of Indicators

Terrain, soil, and vegetation attributes affecting agroecosystem functions were adopted based on literature and EO (Supplementary Table S2). The original eighteen indicators were implied in a total data set (TDS) and were subjected to multivariate statistical analysis using SPSS 19 software. Pearson's correlation analysis was done to test linear relationships (Table 1. Thereafter, factor analysis using PCA with the Kaiser–Meyer–Olkin (KMO) measure of sampling adequacy, Bartlett's test of sphericity, and Varimax rotation was applied to the correlation matrix to build MDS. Only PC with eigenvalues > 1.0 and explaining at least 5% of the data variation were selected [10]. Under each PC, only highly loaded variables with absolute value > 0.6 were retained [17]. Well-correlated parameters were considered redundant, and thus the highest loaded one was only chosen in MDS; meanwhile, highly loaded un-correlated variables were considered important and retained under each PC [8].

**Table 1.** Pearson's correlation matrix of land quality variables.

| Indicator | Slope | Aspect | TWI | SRI | LS | pH | EC | ESP | OM | CaCO3 | Depth | Sand | Silt | Clay | WHC | BD | HC |
|---|---|---|---|---|---|---|---|---|---|---|---|---|---|---|---|---|---|
| Slope | 1.00 | | | | | | | | | | | | | | | | |
| Aspect | 0.34 * | 1.00 | | | | | | | | | | | | | | | |
| TWI | −0.48 ** | −0.23 | 1.00 | | | | | | | | | | | | | | |
| SRI | 0.02 | −0.11 | −0.48 ** | 1.00 | | | | | | | | | | | | | |
| LS | 0.59 ** | 0.20 | 0.14 | −0.47 ** | 1.00 | | | | | | | | | | | | |
| pH | 0.12 | 0.15 | −0.18 | 0.14 | 0.12 | 1.00 | | | | | | | | | | | |
| EC | −0.16 | −0.02 | 0.12 | 0.01 | −0.10 | −0.27 | 1.00 | | | | | | | | | | |
| ESP | −0.25 | −0.03 | 0.23 | −0.15 | −0.07 | −0.13 | 0.82 ** | 1.00 | | | | | | | | | |
| OM | 0.14 | −0.05 | −0.23 | 0.06 | −0.08 | −0.06 | −0.25 | −0.15 | 1.00 | | | | | | | | |
| CaCO3 | −0.41 ** | −0.20 | 0.23 | −0.01 | −0.23 | −0.15 | 0.33 * | 0.26 | −0.14 | 1.00 | | | | | | | |
| Depth | 0.16 | 0.06 | −0.33 * | 0.04 | 0.05 | 0.13 | 0.03 | 0.04 | −0.13 | −0.38 ** | 1.00 | | | | | | |
| Sand | 0.27 | 0.00 | −0.03 | −0.11 | 0.22 | 0.20 | 0.02 | 0.07 | 0.05 | −0.33 * | 0.29 * | 1.00 | | | | | |
| Silt | −0.26 | −0.12 | 0.00 | 0.11 | −0.25 | −0.27 | −0.01 | −0.14 | 0.08 | 0.35 * | −0.19 | −0.88 ** | 1.00 | | | | |
| Clay | −0.23 | 0.10 | 0.06 | 0.08 | −0.15 | −0.02 | −0.02 | 0.00 | −0.16 | 0.26 | −0.33 * | −0.91 ** | 0.62 ** | 1.00 | | | |
| WHC | −0.27 | 0.04 | 0.05 | 0.10 | −0.19 | −0.10 | 0.00 | 0.00 | −0.11 | 0.33 * | −0.33 * | −0.96 ** | 0.74 ** | 0.98 ** | 1.00 | | |
| BD | 0.23 | −0.06 | 0.06 | 0.04 | 0.21 | 0.11 | 0.01 | 0.03 | −0.07 | −0.13 | 0.12 | 0.413 ** | −0.48 ** | −0.27 | −0.35 * | 1.00 | |
| HC | 0.25 | 0.00 | −0.04 | −0.10 | 0.17 | 0.02 | −0.13 | −0.16 | 0.10 | −0.52 ** | 0.28 | 0.80 ** | −0.59 ** | −0.84 ** | −0.88 ** | 0.32 * | 1.00 |
| NDVI | 0.12 | 0.02 | −0.03 | −0.11 | −0.03 | 0.03 | −0.15 | −0.06 | 0.29 * | 0.11 | −0.47 ** | 0.02 | −0.04 | −0.01 | −0.02 | 0.08 | −0.04 |

TWI, topographic wetness index; SRI, surface roughness index; LS, slope-length factor EC, electrical conductivity; ESP, Exchangeable sodium percent; OM, organic matter; WHC, water holding capacity; BD, bulk density; HC, hydrlic conductivity, NDVI, normalized difference vegetation index. Correlation is significant at the 0.05 (*) and 0.01 (**) levels.

### 2.4.2. Generating Thematic Layers

The GIS geostatistical analyst was employed to generate raster maps for soil properties using ordinary kriging (OK). The OK is an advanced technique that predicts the value of a property at an un-sampled point to create continuous layers [34]. The OK is one of the most acceptable methods, which can use a limited set of sampled data points to predict the value of a variable over a continuous spatial field [35]. The predicted value $Z(x_0)$ is estimated using measured data ($Z(xi)$), weights of measured values ($\lambda_i$) within a certain distance, and number of predicted values ($n$) within certain neighbor samples (Equation (5)).

$$Z(x_0) = \sum_{i=1}^{n} \lambda_i \times Z(x_i) \tag{5}$$

Prior performing the OK interpolation, data were explored. The studied soil properties (except pH and OM) did not show normal distribution, and thus transformation using log technique was applied for soil data. Thereafter, the semivariogram was used for fitting the OK models. The semivariogram is a statistic, which assesses the average decrease in similarity between two random variables as the distance between the variables increases, leading to applications in exploratory data analysis [36]. Generally, there are three main properties describing the semivariogram, including nugget effect, range, and sill. The nugget effect represents a discontinuity of the variogram that expresses both variability at a scale smaller than the sampling interval and non-spatial variation. Repeated measurements are the only way to remove the nugget effect that cannot be removed by close sampling [34]. Range and sill express lag distance and distance for uncorrelated samples, respectively [22]. The prediction errors were considered to evaluate and figure out the most suitable model. They included mean error (ME, Equation (6)), mean standardized error (MSE, Equation (7)), and root mean square standardized error (RMSSE, Equation (8)), as follows [34]:

$$ME = \frac{1}{n} \sum_{i=1}^{n} (x_i - y_i) \tag{6}$$

$$MSE = \frac{1}{n} \sum_{i=1}^{n} [x_i - y_1] \tag{7}$$

$$RMSSE = \sqrt{\frac{1}{n} \sum_{i=1}^{n} [x_i - y_i]^2} \tag{8}$$

### 2.4.3. Standardization of Thematic Layers

The GIS-FMFs were applied to transform each pixel (cell) in terrain, soil, and vegetation raster maps into a 0 to 1 scale ($\mu(x)$. The linear-increasing (Equation (9)) and linear-decreasing (Equation (10)) FMFs were applied to represent fuzzy linear scores, while large (Equation (11)) and small (Equation (12)) FMFs were selected to represent fuzzy non-linear scores (Table S2). The linear FMFs generate linear relationships between upper ($U$) and lower ($L$) limits for a parameter ($x$), which are inputted by the user as follows [37]:

$$\mu(x) = \begin{cases} 1 \; if \; x \geq U \\ \frac{x-L}{U-L} \; if \; L < x < U \\ 0 \; if \; x \leq L \end{cases} \tag{9}$$

$$\mu(x) = \begin{cases} 1 \; if \; x \leq L \\ \frac{U-x}{U-L} \; if \; L < x < U \\ 0 \; if \; x \geq U \end{cases} \tag{10}$$

The large FMF is used when larger input values are more preferred to be a member of the set, while the small FMF is used when smaller input values have more membership

values. They depend on spread amounts ($d_1$) and midpoints ($d_2$) set by the user. Formulas showing fuzzy large and fuzzy small are as follows [37]:

$$\mu(x) = \frac{1}{1 + \left(\frac{x}{d_2}\right)^{-d_1}} \tag{11}$$

$$(x) = \frac{1}{1 + \left(\frac{x}{d_2}\right)^{d_1}} \tag{12}$$

### 2.4.4. Weighting Procedure

A weight for each indicator in TDS and MDS was derived from AHP [19]. Opinions of ten local experts and authors' judgments were adopted to prioritize indicators based on their importance. A pairwise comparison matrix (PCM) was built for the main-criteria (terrain, soil chemical, soil physical, and vegetation). The comparison of each criterion to one another was done with a rating scale (1 to 9). Similarly, PCMs were designed for the sub-criteria of each main group and for MDS. Finally, a weight value for each criterion and consistency ratio (CR) for each PCM were estimated (Supplementary Tables S3 and S4) using the AHP software package Expert Choice.

### 2.4.5. Land Quality Index and Classes

The fuzzy maps were compiled using arithmetic mean (A) (Equation (13)) and weighted sum (W) (Equation (14)) models to obtain maps for eight LQIs [11] as follows:

$$LQI_A = \sum_{i=1}^{n} \frac{S_i}{n} \tag{13}$$

$$LQI_W = \sum_{i=1}^{n} Si_i \times W_i \tag{14}$$

where $S_i$ is the indicator score, $W_i$ is the indicator weight, and n is the number of indicators. To select the most appropriate index, the coefficient of variation (CV) and sensitivity index (SI) were considered. The SI was calculated using Equation (15) [38] as follows:

$$SI = \frac{LQI_{maximum}}{LQI_{minum}} \tag{15}$$

The land qualities were arranged in five classes using Jenks's natural breaks, which is the most proper technique to classify uneven distributed data [7]. The quality grades were very high (I), high (II), moderate (III), low (IV), and very low (V).

### 2.5. Developing SLQZs

This implied a further PCA on MDS and then performing HCA. Using PC scores of soil profiles [21], HCA with Ward's linkage method and squared Euclidean distance (SED) as a similarity measure [26] was conducted using SPSS 19 software. One-way ANOVA and Tukey's HSD test were performed to compare the means of generated zones.

### 2.6. Performance Evaluation

The cross-validation was employed to specify the most suitable OK methods. Models with ME and MSE close to zero and RMSSE close to 1.0 were only used [34]. The KMO and Bartlett's tests were implied to test the PCA applicability as values above 0.6 and below 0.05, respectively, were accepted to proceed with PCA [12]. The reality of AHP-weights was checked through CR, where a PCM with CR below 0.10 was accepted, while that of higher values was revised [19]. The coefficient of determination ($R^2$) was computed to measure the similarity between MDS and TDS. The Kappa statistic was conducted to estimate the

agreement between quality grades with limits of [20]: (1) none, <0; (2) slight: 0–0.20, (3) fair: 0.21–0.40; (4) moderate: 0.41–0.60, (5) substantial: 0.61–0.80 and (6) perfect: >0.80.

## 3. Results

### 3.1. Spatial Variability of Soil Attributes

The normality test (Supplementary Table S1) indicates that all the studied soil properties did not show normal distribution, except pH and OM. Therefore, the logarithm transformation method was applied before conducting the OK interpolation techniques (Supplementary Figure S3). The variography analysis of soil properties (Table 2) reveals that the spherical model fitted four properties, i.e., pH, clay, WHC, and HC. Each of the exponential and circular models was proper to map three attributes, including EC, ESP, and sand for the former, while $CaCO_3$, OM, and silt for the latter. The Gaussian model fitted depth and BD. The spatial dependency (SPD) was estimated using the nugget to sill ratio. The SPD is strong, moderate, and weak if this ratio is below 0.25, 0.25–0.75, and above 0.75, respectively [39]. Accordingly, EC, ESP, and $CaCO_3$ showed a strong SPD, silt, depth, WHC, and HC showed moderate SPD, while pH, OM, sand, clay, and BD showed weak SPD. All the applied OK models had ME and MSE close to zero and RMSSE close to 1.0. The spatial distribution maps of soil properties are shown in Supplementary Figure S4.

**Table 2.** Semi-variogram parameters for soil properties.

| Property | Model | Nugget | Sill | Nugget/Sill | SPD | ME | RMSE | MSE | RMSSE | ASE |
|---|---|---|---|---|---|---|---|---|---|---|
| pH | Spherical | 0.08 | 0.09 | 0.88 | Weak | −0.01 | 0.30 | −0.03 | 0.98 | 0.31 |
| EC | Exponential | 0.00 | 0.74 | 0.00 | Strong | −0.05 | 6.30 | −0.06 | 0.77 | 8.80 |
| ESP | Exponential | 0.00 | 0.10 | 0.00 | Strong | 0.00 | 2.88 | −0.02 | 1.04 | 2.72 |
| $CaCO_3$ | Circular | 0.00 | 0.43 | 0.00 | Strong | 0.00 | 14.89 | −0.16 | 1.18 | 21.25 |
| OM | Circular | 0.24 | 0.25 | 0.95 | Weak | 0.01 | 0.51 | 0.02 | 1.01 | 0.50 |
| Sand | Exponential | 0.03 | 0.03 | 0.79 | Weak | 0.08 | 12.82 | 0.00 | 0.89 | 14.24 |
| Silt | Circular | 0.44 | 0.71 | 0.62 | Moderate | −0.07 | 8.02 | −0.01 | 1.07 | 7.43 |
| Clay | Spherical | 0.50 | 0.54 | 0.93 | Weak | −0.04 | 7.39 | −0.01 | 1.01 | 7.32 |
| Depth | Gaussian | 0.01 | 0.02 | 0.47 | Moderate | −0.01 | 13.65 | −0.01 | 0.88 | 15.16 |
| WHC | Spherical | 0.07 | 0.10 | 0.70 | Moderate | −0.06 | 4.81 | −0.03 | 1.20 | 4.23 |
| BD | Gaussian | 0.0526 | 0.06 | 0.82 | Weak | −0.01 | 0.25 | −0.03 | 0.98 | 0.26 |
| HC | Spherical | 0.16 | 0.30 | 0.53 | Moderate | −0.07 | 46.24 | 0.00 | 1.01 | 45.65 |

EC, electrical conductivity; ESP, exchangeable sodium percentage; OM, organic matter; WHC, water holding capacity; BD, bulk density; HC, hydraulic conductivity; SPD, spatial dependency; ME, mean error; RMSE, root mean standardized error; MSE, mean square error; RMSSE, root mean square standardized error; ASE, average standardized error.

### 3.2. Multivariate Statistical Analysis

#### 3.2.1. Correlation Analysis

The slope had a significant ($p < 0.05$) positive correlation with aspect and a highly significant ($p < 0.01$) positive correlation with LS-factor (Table 2). However, the slope was negatively associated with TWI and $CaCO_3$. The TWI was negatively correlated with SRI and soil depth. A highly significant negative correlation occurred between SRI and LS-factor. The EC was positively correlated with ESP and $CaCO_3$. The OM showed a significant positive correlation with NDVI. The $CaCO_3$ was positively correlated with silt and WHC, but negatively associated with depth, sand, and HC. The depth was positively correlated with sand but negatively associated with clay, WHC, and NDVI. The sand showed positive correlations with BD and HC, but negative correlations with silt, clay, and WHC. A highly significant positive correlation occurred between silt and clay. They were positively correlated with WHC, but negatively correlated with HC. The WHC was negatively correlated with BD and HC. There was a significant positive correlation between BD and HC.

### 3.2.2. Principal Component Analysis

The PCA (Table 3) illustrates that the KMO and significance of Bartlett's test were 0.66 and zero, respectively. The first six PCs had eigenvalues above 1, which explained 77.27% of the total variance. The PC1 represented 26.39% of the total variance, including sand, and HC with high positive loadings and silt, clay, and WHC with high negative loadings. Terrain attributes (slope, aspect, and LS-factor) were correlated under PC2, explaining 12.06% of the total variance The EC and ESP dominated PC3, representing 11.72% of the total variance. The PC4 explained 10.56% of the total variance, including SRI (high positive loading) and TWI (high negative loading). The PC 5 represented 8.85% of the total variance, including NDVI (high positive loading) and soil depth (high negative loading). The PC6 with a high positive loading of pH and high negative loading of OM exhibited 7.70% of the total variance.

**Table 3.** Principal component analysis of the studied indicators.

| Parameter | PC1 | PC2 | PC3 | PC4 | PC5 | PC6 |
|---|---|---|---|---|---|---|
| Eigenvalue | 4.751 | 2.170 | 2.109 | 1.900 | 1.593 | 1.385 |
| Variance, % | 26.392 | 12.057 | 11.717 | 10.557 | 8.849 | 7.696 |
| Cumulative, % | 26.392 | 38.449 | 50.166 | 60.723 | 69.573 | 77.269 |
| Indicator | | | Eigenvector | | | |
| Slope | 0.234 | **0.825** | −0.149 | 0.096 | 0.068 | −0.018 |
| Aspect | −0.115 | **0.697** | 0.049 | 0.006 | −0.063 | 0.103 |
| TWI | −0.027 | −0.454 | 0.111 | **−0.784** | 0.065 | 0.158 |
| SRI | −0.092 | −0.169 | −0.035 | **0.826** | −0.076 | 0.190 |
| LS | 0.169 | **0.631** | −0.086 | −0.566 | 0.008 | 0.134 |
| pH | 0.100 | 0.170 | −0.234 | 0.260 | 0.027 | **0.703** |
| EC | −0.004 | −0.052 | 0.951 | 0.004 | −0.092 | −0.049 |
| ESP | 0.035 | −0.060 | 0.921 | −0.090 | −0.011 | 0.012 |
| OM | 0.154 | 0.148 | −0.224 | 0.366 | 0.263 | **−0.613** |
| CaCO$_3$ | −0.353 | −0.450 | 0.383 | −0.047 | 0.351 | 0.022 |
| Depth | 0.308 | 0.207 | 0.021 | 0.227 | **−0.714** | 0.111 |
| Sand | **0.969** | 0.079 | 0.049 | −0.016 | −0.021 | 0.138 |
| Silt | **−0.786** | −0.178 | −0.092 | 0.034 | −0.041 | −0.428 |
| Clay | **−0.945** | 0.022 | −0.003 | −0.003 | 0.071 | 0.141 |
| WHC | **−0.981** | −0.035 | 0.010 | 0.010 | 0.057 | 0.023 |
| SBD | 0.448 | 0.137 | 0.065 | −0.049 | 0.150 | 0.408 |
| HC | **0.878** | 0.045 | −0.195 | −0.063 | −0.156 | −0.091 |
| NDVI | 0.069 | 0.093 | −0.078 | 0.052 | **0.898** | −0.112 |
| Kaiser–Meyer–Olkin (KMO) and Bartlett's statistics | | | | | | |
| KMO Measure of Sampling Adequacy | | | | | | 0.664 |
| Bartlett's Test of Sphericity | | | | Approx. chi-square | | 1246.58 |
| | | | | Degree of freedom | | 153 |
| | | | | Significance | | 0.000 |

Bold face numbers indicate highly-loaded variables.

The dendrogram (Figure 2) shows results of HCA of soil profiles. The visual interpretation reveals that they could be grouped in eight clusters, reflecting SLQZs. The SLQZ1 had the highest number of soil profiles, representing 36% of the total data. The SLQZ2 and SLQZ7 involved the lowest number of soil profiles, and each of them represented only 2% of the total data. The statistical analysis indicates significant differences between all attributes among the eight zones (Supplementary Table S5). Considering salinity as a major limiting factor, the SLQZ2 and SLQZ7 with the lowest and highest EC values, respectively, would have higher and lower potential quality than others. The remaining zones would have moderately high (zones 1, 2, 3, 5 and 8) and moderately low (zone 6) potential qualities.

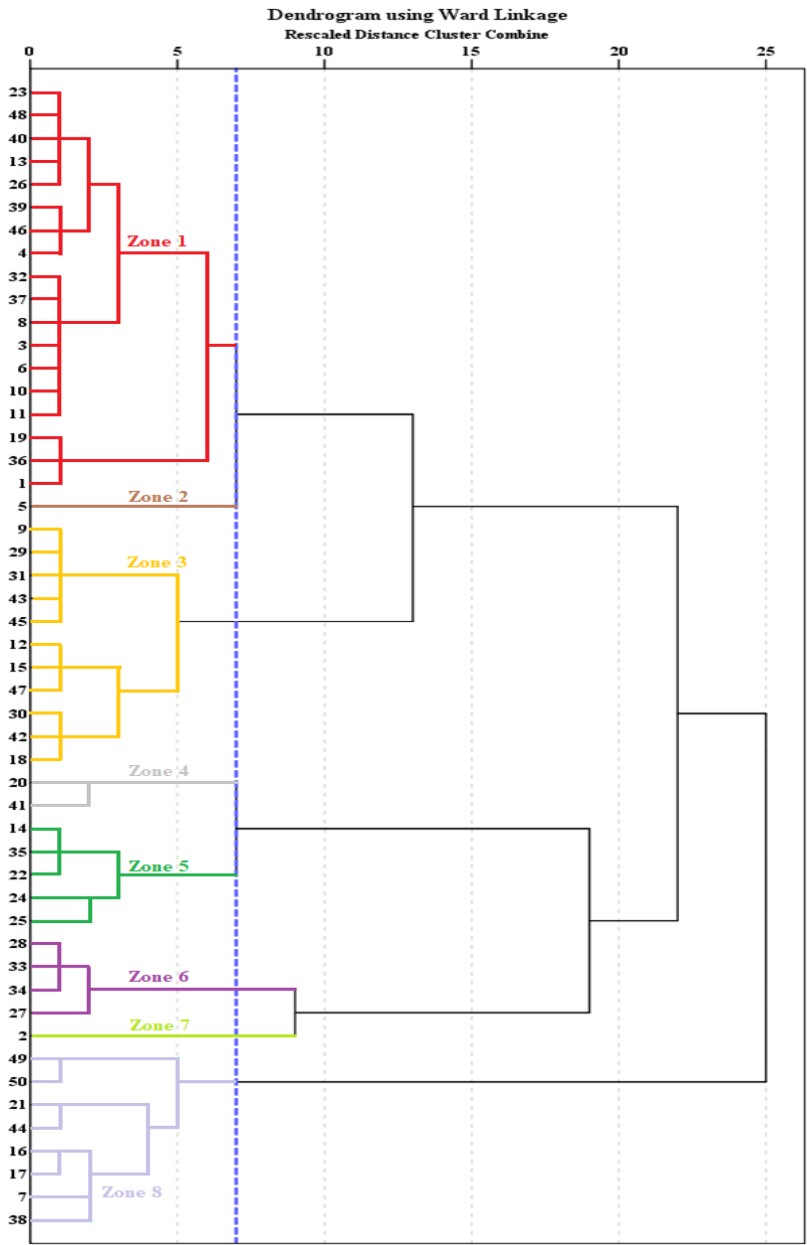

**Figure 2.** Dendrogram for the soil profile clusters in the studied area.

### 3.3. Land Quality Assessment

3.3.1. According to TDS

Maps of the developed LQIs are shown in Figure 3 and classifications of quality grades are listed in Supplementary Table S6. The LQIA generated from the linearly scored TDS revealed that areas fitted grades I, II, III, IV, and V covered 27, 33, 22, 11, and 7% of the total area, respectively. However, using the non-linear scores, areas fitted the same classes occupied 18, 20, 21, 26, and 15% of the total area, respectively. Of the original 18 indicators, soil chemical properties (EC, ESP, CaCO$_3$, pH, and OM) had higher specific weights than soil physical (depth, clay, WHC, sand, BD, and HC) and terrain attributes (slope, SRI, LS-factor, TWI, and aspect), while NDVI had the lowest weight (Supplementary Table S3). According to the LQI$_W$ developed from the linear scores, areas in grades I and II occupied 77% of the studied area, while those in grades III, IV, and V covered 11, 8, and 4%, respectively. Using the non-linear scores, 17, 27, 25, 20, and 11% of the total area occurred in grades I, II, III, IV, and V, respectively.

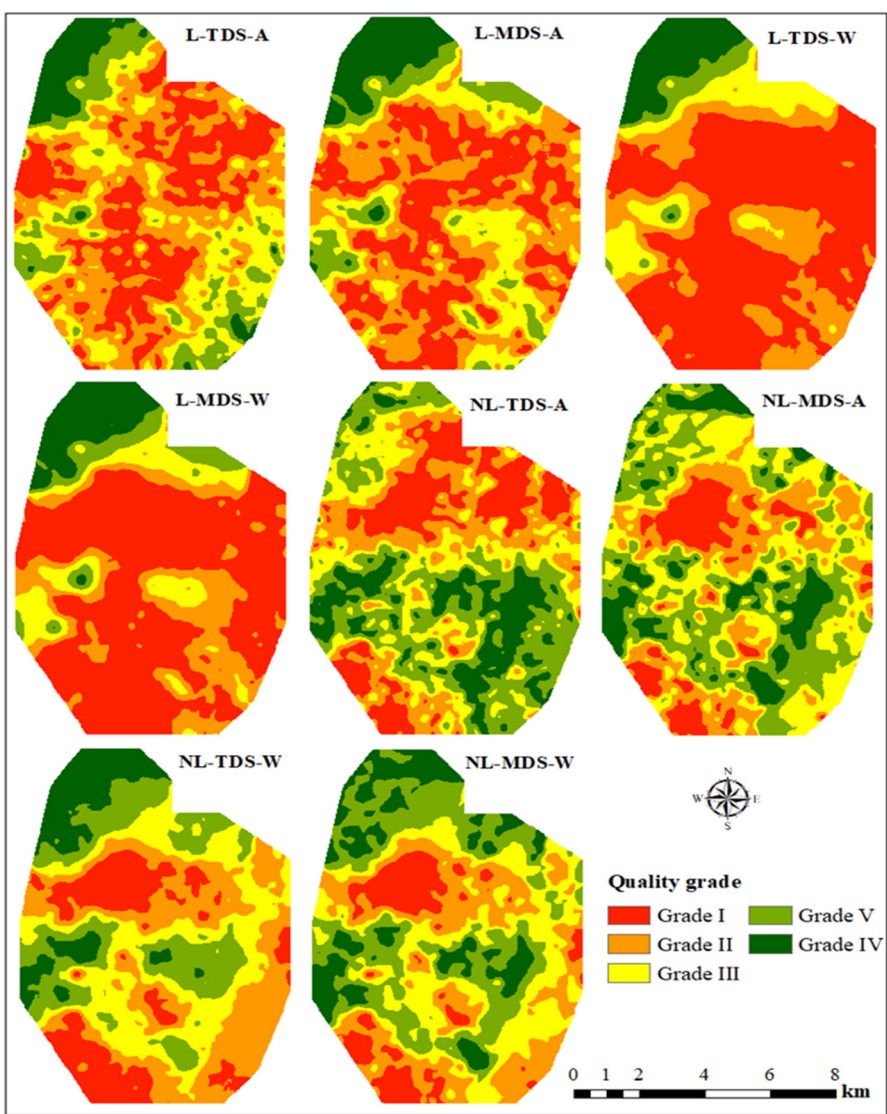

**Figure 3.** Maps of the developed land quality indices using linearly (L) and non-linearly (NL) scored total data set (TDS) and minimum data set (MDS) under arithmetic mean (A) and weighted sum (W) models.

### 3.3.2. According to MDS

The PCA revealed that only seven indicators were adequate to model LQIs, including sand, slope, EC, SRI, NDVI, pH, and OM. The LQI$_A$ based on the linearly scored MDS indicates that areas in grades I and II accounted equally for 32% of the total area, areas of grades IV and V were equally distributed in 9%, while areas in grade III covered 18%. However, based on the non-linear scores, 11, 19, 29, 28, and 12% of the total area were in grades I, II, III, IV, and V, respectively. The EC had the highest weight (0.36), followed by slope (0.19), pH (0.12), SRI (0.11), NDVI (0.09), and OM (0.08), while sand had the lowest weight (0.06) (Supplementary Table S4). The LQIW based on the linearly scored MDS shows that 75% of the total area was in grades I and II, while areas in grades III, IV, and V covered 11, 5, and 9%, respectively. The spatial distribution of quality grades using the non-linear scores was as follows: 11% for grade I, 23% for grade II, 25% for grade III, 27% for grade IV, and 15% for grade V.

### 3.4. Comparison of Indices

As shown in Figure 4, there were significant differences ($p < 0.05$) among the LQIs. However, correlation (Supplementary Table S7) shows that they were all significantly positively correlated. The linear regression (Figure 5) reveals that LQIs (LQIA and LQIW) generated from TDS and MDS were highly correlated with one another. Regarding the agreement between quality grades set by TDS and MDS, Kappa statistics under the LQIA were 0.33 (fair) and 0.68 (substantial) using the linearly and non-linearly scored indicators, respectively. These values under the LQIW were 0.39 (fair) and 0.62 (substantial) using the linearly and non-linearly scored indicators, respectively. The highest SI (3.88) and CV (35.43%) values occurred for the LQIW from the non-linearly scored MDS (Figure 4). On the other hand, the lowest SI (1.17) and CV (4.04%) values occurred for the LQIA from the linearly scored TDS.

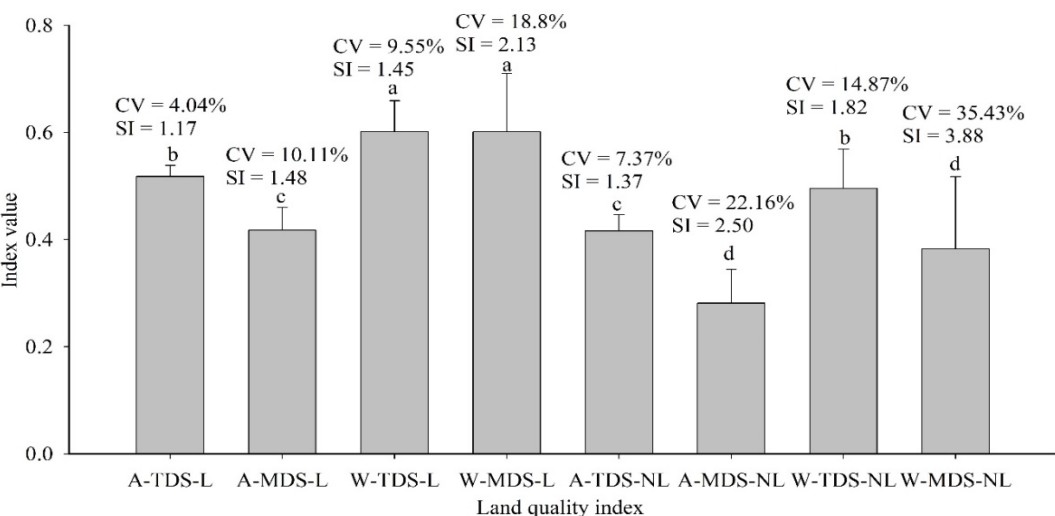

**Figure 4.** Comparisons of land quality indices, through calculating coefficient of variation (CV) and sensitivity index (SI).

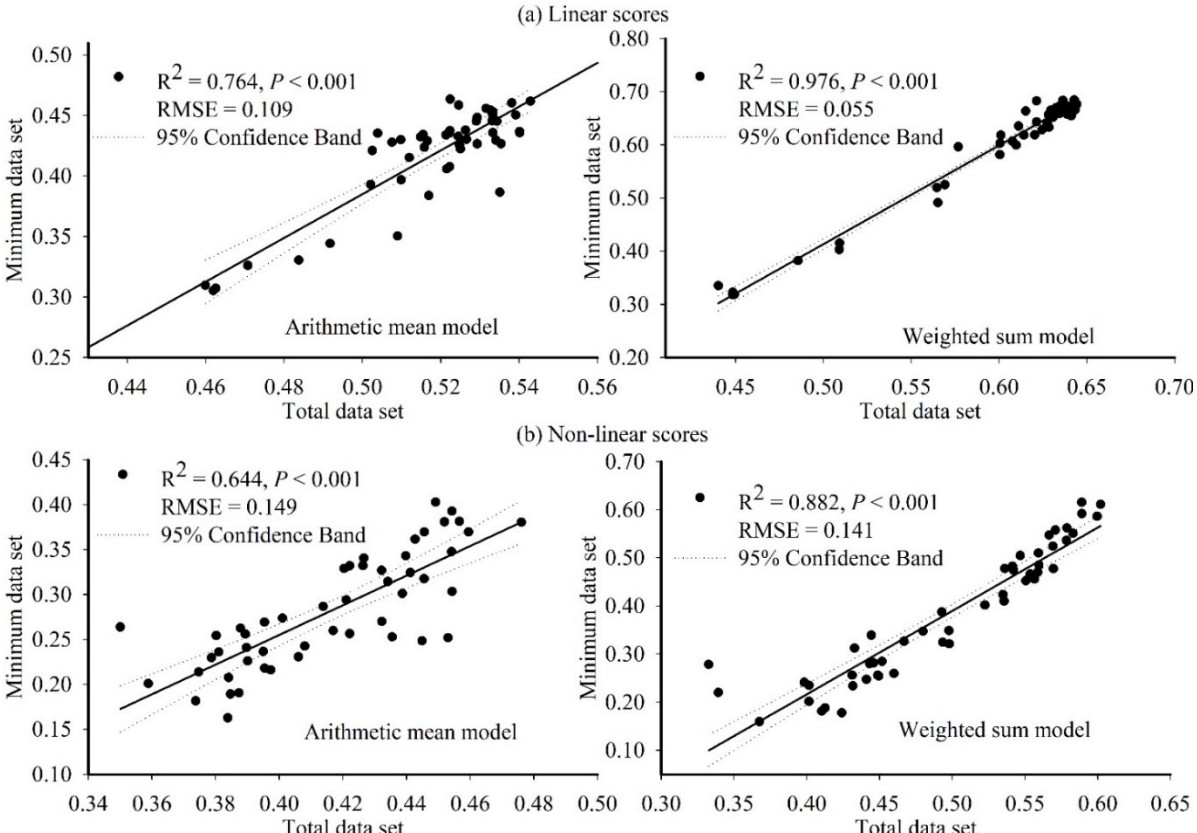

**Figure 5.** Relationships between indices developed from the total data set and the minimum data set.

## 4. Discussion

### 4.1. Spatial Variability of Soil Attributes

The OK models provided reliable predictions for soil attributes, confirming the findings of Abdellatif et al. [20], who indicated the reality of OK models (spherical, exponential, circular, and Gaussian) in mapping soil properties in west Matrouh Governorate. Generally, the best semivariogram models are usually chosen based on prediction errors, including ME, MSE, RMSSE, and SPD. Models with ME and MSE close to zero provide unbiased estimation for un-sampled locations, and RMSSE close to 1.0 confirms an accurate prediction and reveals that the quality and appropriateness of the predicting model are high [34]. The SPD revealed different types of soil heterogeneity due to various factors [40,41]. The strong SPD may be due to natural factors, while the weak and moderate may be attributable to other factors, for instance, unsuitable agricultural practices and agricultural management [39]. As 97% of the studied area is un-cultivated, intrinsic factors were mainly responsible for soil spatial variability [24]. The strong heterogeneity of EC, ESP, and $CaCO_3$ suggests that they were controlled by parent materials and soil-forming processes [39]. The slope gradient might affect the lime distribution in soils, and this trend was depicted by the significant correlation between slope and $CaCO_3$. The fine-scaled soil discontinuities caused by inherent variations of soil quality (texture and mineralogy) might lead to the small-scale heterogeneity of the remaining attributes [24].

### 4.2. Relationships among Land Quality Attributes

#### 4.2.1. Correlation Analysis

The correlation reflected effects of terrain and soil attributes on hydrological conditions. Flat and rough surfaces improve soil moisture distribution by enhancing vertical water flows and reducing horizontal runoff [30]. Shallow soils with high lime content could also support moisture retention. On steep slope areas, loess calcareous materials derived from limestone are eroded in response to Aeolian processes [2]. Long-slope faces in the

studied area were covered by coarse fragments that raised the SRI. The EC relations with ESP and CaCO$_3$ prove that soil salinity was due to Na$^+$, Ca$^{2+}$, CO$_3^{2-}$ and HCO$_3^-$ [42]. The increased vegetation might promote biomass production and support soil OM content. The particle size distribution of CaCO$_3$ has key effects on soil properties [43]. The CaCO$_3$ in the studied soils might be in the silt and clay-sized fractions, causing positive effects via enhancing WHC and reducing HC. However, excessive lime content led to hardpan formations that lowered soil depth. This might enhance soil water content and increase the NDVI. The fine-earth had key effects on soil quality since soils originated from coarse sands are deep with poor physical properties like high density, rapid infiltration rate, and low water retention [44]. The increments of medium (silt) and fine (clay) particles might ameliorate such conditions since these fractions block soil macro-pores, decreasing BD and HC, and improving WHC [45].

### 4.2.2. Principal Component Analysis

The linear correlations among the variables reveal that the PCA would perform well [10,46]. This assumption was supported by the KMO statistic (>0.6) and Bartlett's test (<0.05) [12]. The first PC could denote soil physical properties governing hydrological conditions, explaining the greatest variation. The terrain attributes dominated PC2 and PC4 and explained together 22.61% of the total variance. The PC2 could depict the effect of slope length and aspect, while the PC4 could reflect the contribution of terrain roughness to moisture retention. The soil chemical properties explained 20.57% of the total variance and dominated PC3 and PC6. The PC3 could verify the dual effect of salinity and sodicity, while PC6 could affirm the positive impact of OM on soil pH. The PC5 could affirm the negative effect of soil depth on biomass production and vegetation cover.

### 4.2.3. Cluster Analysis

The cluster analysis allows discriminating various zones with a similar value of characteristics and higher variations between them. Classifications with K-means [21] and HCA [25,26] have been applied to zone fields into specific units. However, HCA has been approved as the most effective technique for this goal [25,26]. It is an unsupervised model, in which samples are successively clustered into a distance matrix computed from the data to draw a dendrogram depicting the groups [26]. The clusters are delineated by setting a phenon line across the dendrogram. In this work, the dendrogram was supported by a phenon line at a SED of 7, and thus samples below this line were in the same cluster. Hence, soils occurred in eight SLQZs that showed significant differences in all attributes. This confirms the reality of HCA in zoning the studied area into various parts [21,26].

Each zone had similar soil properties that impose certain management practices. The worst salinity level was in SLQZ7 (strongly saline) that declined in SLQZ6 (moderately saline). Adaptation to salinity stress entails selecting salt-tolerant crops, applying excess water to leach soluble salt, and adopting sulfur and organic additions [42]. Slightly (zone 1, 2 and 5) and very slightly (zone 3 and 8) saline soils can be cultivated by moderately tolerant and moderately sensitive crops, respectively, besides applying leaching fractions to prevent more salinity development [42]. The SLQZ4 had no salinity hazards; however, the highest surface irregularity occurred in this zone. Hence, land leveling can improve surface irrigation and drainage and render the area more manageable for farming practices. For all zones, micro-irrigation (sprinkler and drip systems) is recommended. Compared with surface methods, micro-irrigation achieves higher water and fertilizers use efficiencies and crop yield [43]. It also enables the control of problems of salinity, sodicity, lime, coarse texture, and sloping surfaces [13].

### 4.3. Land Quality Assessment

#### 4.3.1. According to TDS

The indicators adopted in TDS have been implied in previous works to simulate agroecosystem dynamics in Egypt [3,20,47] and other drylands in Iran [48] and Nigeria [49]. These indicators had unequal importance for the LQI, as indicated by AHP-weights. The EC, ESP, and soil depth had the highest impacts, representing 42% of the total weights. The EO could be accepted as CRs for PCMs did not exceed the critical limit (0.1), indicating consistent judgments [19]. Low rainfall and high lime content render salinity, alkalinity, and depth development major threats for dryland agriculture since they lead to irreversible changes in ecosystem services [3,42].

#### 4.3.2. According to MDS

Out of the original eighteen parameters, only seven were selected as key indicators in MDS. The selection of MDS relied on PCA and correlation results, which have been indicated as the most proper reduction techniques for environmental data [8,10,11]. The PC1 was dominated by five indicators; however, they were highly correlated to each other, and thus the sand was informative for this PC [8]. Similarly, slope, EC, SRI, and NVDI represented PC2, PC3, PC4, and PC5, respectively. On the other hand, the PC6 was dominated by pH and OM; however, they were not significantly correlated. Thus, they were included in MDS to explain this PC [8]. The AHP with an acceptable PCM (Table 3) for the seven indicators (CR = 0.049) indicated that EC had the highest specific weight (0.36), while sand had the lowest effect (0.06).

#### 4.3.3. Land Quality Grades

The Jenks algorithm arranged the studied area into five quality grades with different spatial distributions among LQIs. These findings are similar to those of Nabiollahi et al. [50] from Iran, who used diverse indicators, scores, and models, and found varied portions of quality grades. This could explain the unequal performance of the implemented methods [15]. Using the eight LQIs, the studied area was in three quality levels: high (grades I and II), moderate (grade III), and poor (grades IV and V). These levels reflected dominant limitations for ecosystem functions [51]. High-quality areas occupied 4211 ha (53%) in the northern, northeastern, and southern zones, where the lowest limitations occurred. The soils were very deep (depth > 150 cm), non-saline to moderately saline (EC 0.61 to 6.72 dS m$^{-1}$), non-sodic (ESP 4.44 to 9.02), and slightly to moderately calcareous (CaCO$_3$ 1.96 to 74.70 g kg$^{-1}$). Moderate-quality areas occurred all over the area (except the southeastern and southwestern parts), covering 1589 ha (20%). Increased limitations were detected for soil depth (70–150 cm), salinity (EC 4.11 to 10.64 dS m$^{-1}$), sodicity (ESP 7.57 to 17.35) and CaCO$_3$ content (7.33 to 82.83 g kg$^{-1}$). Poor-quality areas covered 2145 ha (27%) in the northwestern parts and scattered areas in the northeast, middle and southern zones. Therein, the worst conditions were observed as the highest values of EC (51.52 dS m$^{-1}$), ESP (28.30), and CaCO$_3$ (129.42 g kg$^{-1}$) occurred.

### 4.4. Comparison of Indices

The applied datasets, FMFs and models caused significant variations among LQIs. However, the eight indices were positively correlated to one another, indicating that each of them can be adopted to track changes in land quality. These results are consistent with those of Zhou et al. [52] from China and Mamehpour et al. [15] from Iran, who found significant variations but positive correlations among indices obtained from diverse methods. The linear regression revealed that MDS could adequately represent TDS under all scores and models. This confirms the reality of PCA in developing science-based, cost-effective, and time-saving LQIs. In many drylands, key indicators derived from the PCA have been successfully applied to monitor agroecosystem functions under croplands in Egypt [53] and India [9], or under pasture in Turkey [7].

Using the linearly and non-linearly scored indicators, values of $R^2$ and Kappa statistics under the $LQI_W$ were higher than those under the $LQI_A$. These results are similar to those of Nabiollahi et al. [50] and Saleh et al. [47], who applied various indices for agricultural lands in Iran and Egypt, respectively, and found that weighted models outperformed additive indices. Vasu et al. [9] reported that weighted models for soil quality in the Deccan plateau, India, showed better correlations with crop yields than additive models. This trend could be due to weights that specify the relative contribution of each property independently [53]. In contrast, the arithmetic mean algorithm computes the summation of scores without weights, and thus all factors have the same effect. This leads to uncertainty since indicators differ in their ability to affect land performance and crop yield [9].

Under $LQI_A$ and $LQI_W$, the similarity between the linearly scored MDS and TDS surpassed those between non-linearly scored datasets. This might reflect the simple calculations used for linear scores, causing higher consistency between the results [52]. However, the non-linearly scored datasets yielded more variations in the LOIs than linearly scored ones did, implying that non-linear FMFs were preferred to quantify ecosystem functions. This could be depicted by higher SI and CV values for LQIs developed from the non-linearly scored datasets than those from linear scores. These findings are rather similar to those reported in earlier works for other agricultural drylands in India [38] and Iran [15], confirming that indices calculated from non-linear scores had higher SI than those from linear scores. The non-linear scoring provides a deeper view of how each indicator affects the ecosystem functions [15].

Overall, the $LQI_W$ based on non-linearly scored MDS could be the most proper index to model land quality in the studied area. Recently, linearly and non-linearly scored MDS weighted by PCA have been adopted in weighted indices to assess dryland agroecosystems. In global studies [15,38], the non-linear scores had better performance than linear systems. However, the linear scores were superior to non-linear scores under Egyptian conditions [47]. In the current work, due to limited data, AHP-weights were applied to compute weighted indices to reduce uncertainty and increase the reality [3,10]. The $LQI_W$ based on the non-linearly scored MDS had the highest SI, rendering it more sensitive to potential changes related to natural and/or human processes [38]. This index also showed the heights CV, indicating the better differentiation ability in the land quality assessment [52]. The MDS used in this index could adequately represent 88% of the variation in TDS, reducing the time, effort, and cost involved in evaluating land quality [50].

## 5. Conclusions

This work provided a novel trail to integrate multi-indicator analysis methods (PCA, HCA, and AHP) with geospatial techniques (RS and GIS) to zone and map arable land quality. The procedures were adopted for an arid desert area (Matruh Governorate) typical for dryland agroecosystems. The PCA could adequately shortlist eighteen indicators for terrain, soil, and vegetation attributes to only seven; these included EC, slope, pH, SRI, NVDI, OM, and sand. Accordingly, the optimum number of SLQZs delineated through PCA and HCA was eight with significant heterogeneity. This, in turn, allowed a policy to be suggested to select proper crops, apply leaching fraction, adopt sulfur and organic applications, perform land leveling, and use micro-irrigation. The AHP could provide real weights for indicators in TDS and MDS. The GIS-spatial analysis and fuzzy tools could efficiently manipulate inputted data to develop eight LQI maps. Each of the generated indices would be effectively applied to monitor land quality dynamics. Nevertheless, the index developed from the non-linearly scored MDS and weighted sum model was the most proper index. It had the highest discrimination ability in land quality and sensitivity to potential natural and human interventions. The MDS in this index could sufficiently represent TDS, and thus reduce the time, effort, and cost for assessing land quality. The proposed methodology would be a start point for precise site-specific management in similar regions. It would also be applied effectively to monitor temporal changes in land quality under agricultural uses. It also reveals that a combination of statistical and

geostatistical approaches is effective and easy to implement in the regions where data availability is constrained. However, increasing the sampling size is recommended for improving the spatial interpretation of land quality indicators.

**Supplementary Materials:** The following are available online at https://www.mdpi.com/article/10.3390/su14105840/s1, Figure S1: Vegetation and terrain attributes of the studied area; Figure S2: Schematic diagram of the methodology adopted in this work, Figure S3: Semi-variogram of OK modelsand Figure S4: Spatial distribution maps of soil properties, Table S1: Descriptive statistics of main soil properties; Table S2: Land quality in-dicators and their linear (L) and non-linear (NL) fuzzy membership functions (FMFs); Table S3: Weights of indicators included in the total data set; Table S4: Pairwise com-parisons and weights of the minimum data set criteria; Table S5: Mean values of proper-ties in the developed spatial land quality zones (SLQZs); Table S6: Spatial distributions of land quality grades according to different indices and Table S7: Pearson's correlation matrix for land quality indices (LQIs). References [54,55] are cited in the Supplementary Materials.

**Author Contributions:** Conceptualization, A.S.A., M.S.S. and A.A.E.B.; methodology, A.S.A., M.S.S. and A.A.E.B.; software, A.S.A., M.S.S. and A.A.E.B.; validation, A.S.A., M.S.S. and A.A.E.B.; formal analysis, A.S.A., M.S.S., Y.S.A.M. and A.A.E.B.; investigation, A.S.A., M.S.S. and A.A.E.B.; resources, A.S.A., M.S.S. and A.A.E.B.; data curation, A.S.A., M.S.S. and A.A.E.B.; writing—original draft preparation, A.S.A., M.S.S. and A.A.E.B.; writing—review and editing, A.S.A., M.S.S., Y.S.A.M., Z.D. and A.A.E.B.; visualization, A.S.A., M.S.S. and A.A.E.B.; supervision, A.S.A., M.S.S. and A.A.E.B.; project administration, A.S.A., Y.S.A.M. and Z.D.; funding acquisition, Y.S.A.M. and Z.D. All authors have read and agreed to the published version of the manuscript.

**Funding:** King Khalid University for funding this work, through Program of Research Groups under grant number (RGP 2/67/43).

**Institutional Review Board Statement:** Not applicable.

**Informed Consent Statement:** Not applicable.

**Data Availability Statement:** Not applicable.

**Acknowledgments:** The authors extend their appreciation to the Deanship of Scientific Research at King Khalid University for funding this work through Program of Research Groups under grant number (RGP 2/67/43). The authors are grateful to Senior Foreign Expert Project of China (Grant number G2021034008L).

**Conflicts of Interest:** The authors declare no conflict of interest.

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
