# Peer review of "Multi-Indicator and Geospatial Based Approaches for Assessing Variation of Land Quality in Arid Agroecosystems"

_sustainability, doi:10.3390/su14105840_

Round 1
Reviewer 1 Report
This study was done based on soil sample data, which are detailed, but overall lack novelty.
Although not novel, it can be considered an annual analysis of Egyptian Western Desert soil properties.
The summary and discussion sections only have the results of the soil correlation study and lack description of how this study could be a guidance for specific practice activities. If this study is intended to guide real-world practice, it should indicate how the results of the study address the problems of practice. Rather than a mere correlation analysis among indicators.
Instead of using the phrase "can guide precision agriculture" . I recommend the author to add the meaning of each indicators and how these indicators changed by farming practice and how study can guide the practice in detail.
Author Response
Many thanks for your valuable comments, all responses in red colour
- This study was done based on soil sample data, which are detailed, but overall lack novelty. Although not novel, it can be considered an annual analysis of Egyptian Western Desert soil properties.
Response: in the drylands, limited studies focused on testing combinations of PCA, AHP, GIS-fuzzy sets to assess arable land resources and establish SLQZs through HCA and The proposed methodology would be a start point for precise site-specific management in similar regions. It would also be applied effectively to monitor temporal changes in land quality under agricultural uses.
- The summary and discussion sections only have the results of the soil correlation study and lack description of how this study could be a guidance for specific practice activities
Response: The recommended management practices have been added in the abstract section
- If this study is intended to guide real-world practice, it should indicate how the results of the study address the problems of practice. Rather than a mere correlation analysis among indicators.
Response:The problems of practices have been addressed in the discussion section (Lines 257 to 269)
- I recommend the author to add the meaning of each indicators and how these indicators changed by farming practice and how study can guide the practice in detail. Response: As we have a lot of analysis in this paper and to be more accurate and concise, we add just the limiting factor in each zone and how to solve the problems related to these factors as we think it important for decision makers so we did not the meaning of each indictor but we will consider it in the future research.

Reviewer 2 Report
Dear Authors,
The main advantage of the manuscript is the use of the multi-indicator approach in soil research and its application to arid agrosystems. However, the main weakness of the article is the lack of in-depth analysis in particular including precise error analysis. In fact, using GIS standard software, several relatively easy-to-calculate indicators were generated and compiled. The work lacks experimental variograms and a detailed description of their modeling. The parameters listed in Table 1 are not sufficient to assess the quality of the modeling as the variogram also depends on the model choice, and this choice can only be assessed when experimental variograms are presented. In general, variograms can be very different, also complex (e.g. Mining geostatistics. Journel, A. G., & Huijbregts, C. J. 1978. Academic Press London; New York; Estimation of soil salinity in semi-arid land using a geostatistical model, 2007. Jose Navarro- Pedreñoet al., Land Degradation and Development 18 (3): 339-353; Geostatistical evaluation of lead and zinc concentration in soils of an old mining area with complex land management, Zawadzki J., FabijaÅ„czyk P. International Journal of Environmental Science and Technology, 2013, 10 (4), 729-742. Besides, the ordinary kriging method cannot be considered (always) the optimal method. Please, mention the above at least in the discussion and supplement the bibliography. The choice of the measurement network has not been fully explained. 50 fairly evenly distributed points are not always enough for the precise determination of the variogram. In particular, in order to accurately determine the nugget effect, it would be necessary to take some measurements placed close to each other. (I am aware of possible measurement limitations). In the discussion, these issues were mentioned (e.g. " Abdellatif et al. [19]). Authors should also better describe the weaker sides of the work, especially in the conclusions.
Minor remarks
There are many errors and typos in the text and even in the formulas. For example, (line 170) incorrect index in Z (x0); (line 180) no square in formula (7); (line 510 double dot; in Formula 8, the right-hand expression is abbreviated as RMES and in the text is mentioned as RMSSE., etc., etc.
Best regards,
Reviewer
Author Response
Many thanks for your valuable comments, all responses in red colour
- The work lacks experimental variograms and a detailed description of their modeling
.Response: the experimental variograms and their detailed description of their modeling have been added in the result section and supplementary figures
- The parameters listed in Table 1 are not sufficient to assess the quality of the modeling as the variogram also depends on the model choice, and this choice can only be assessed when experimental variograms are presented.
Response: a lot of research papers used these parameters for assessing and model land quality in arid zones and gave sufficient results such as,.
Abdellatif, M.A.; El Baroudy, A.A.; Arshad, M.; Mahmoud, E.K.; Saleh, A.M.; Moghanm, F.S.; Shaltout, K.H.; Eid, E.M.; Shokr, M.S. A GIS-Based Approach for the Quantitative Assessment of Soil Quality and Sustainable Agriculture. Sustainability 2021, 13, 13438. https://doi.org/10.3390/su132313438
Abdel-Fattah, M.K.; Mohamed, E.S.; Wagdi, E.M.; Shahin, S.A.; Aldosari, A.A.; Lasaponara, R.; Alnaimy, M.A. Quantitative Evaluation of Soil Quality Using Principal Component Analysis: The Case Study of El-Fayoum Depression Egypt. Sustainability 2021, 13, 1824. https://doi.org/10.3390/su13041824
- Please, mention the above at least in the discussion and supplement the bibliography.
Response: the recommended references have been added to the discussion and bibliography
- The choice of the measurement network has not been fully explained.
Response: Wwe choose the measurements points randomly to represent the geomorphologic units of study area and it is clear from geologic ma that there are no big differences in geologic units within study area
50 fairly evenly distributed points are not always enough for the precise determination of the variogram
Response: we agree but, we choose the maximum points that represent the study area as this area is desert and there a lot of measurement limitations but
- In particular, in order to accurately determine the nugget effect, it would be necessary to take some measurements placed close to each other. (I am aware of possible measurement limitations).
Response: we agree but there are a lot measurement limitations and we will take it in our consideration in the near future publications
- Authors should also better describe the weaker sides of the work, especially in the conclusions.
Response: the weakness points have been mentioned in the conclusion
- There are many errors and typos in the text and even in the formulas. For example, (line 170) incorrect index in Z (x0); (line 180) no square in formula (7); (line 510 double dot; in Formula 8, the right-hand expression is abbreviated as RMES and in the text is mentioned as RMSSE., etc., etc.
Response: the typos have been corrected and the manuscript have been revised
